# Successes and challenges of an online based nutrition awareness program in 9–11-year-old children in four Arab countries: The Ajyal Salima digital platform qualitative study

Carla Habib-Mourad[1]*, Carla Maliha[1], Amira Kassis[2], Diala Tailfeathers[3], Marco Bardus[4], Eman Haji[5], Lina AlTarazi[6], Suzan Totah[7], Nahla Hwalla[1]

**1** Department of Nutrition and Food Sciences, Faculty of Agriculture and Food Sciences, American University of Beirut, Beirut, Lebanon, **2** Neat Science, Châtel-Saint-Denis, Fribourg, Switzerland, **3** Department of health and Physical Education, Mount Royal University, Calgary, Canada, **4** Department of Applied Health Sciences, School of Health Sciences, College of Medicine and Health, University of Birmingham, Birmingham, United Kingdom, **5** Ministry of Health, Sanabis, Kingdom of Bahrain, **6** Royal Health Awareness Society, Amman, Jordan, **7** Ministry of Education, Ramallah, Palestine

* ch18@aub.edu.lb

## Abstract

### Introduction

The rapid expansion of digital technologies has significantly influenced the lives of children and youth, leading many to seek nutrition education through digital platforms. This study aims to assess the usability and acceptability of Ajyal Salima, a nutrition awareness digital platform targeting children aged 9–11, in four Arab countries.

### Methods

A qualitative study was led across four countries: Lebanon, Bahrain, Palestine, and Jordan. Semi-structured focus groups discussions (FGDs) were held separately with children (21 FGDs; n = 145) and parents (16 FGDs; n = 98), complemented by In Depth Interview (IDIs) with teachers (n = 19) and Key Informant Interviews (KIIs) with program staff (n = 8). All interviews and focus groups lasted approximately 40 minutes. Data was analyzed thematically using NVivo software, resulting in four main themes.

### Results

Four major themes emerged: platform's usability, content enjoyment, changes in children's habits and recommendations to improve the platform. Overall, parents and teachers found the digital experience positive and useful and the content appropriate for children, particularly younger age groups. Challenges included registration difficulties, technical problems, internet accessibility, low parental involvement, and

**Data availability statement:** All relevant data are within the manuscript and its Supporting Information files.

**Funding:** Nestle For Healthier Kids Initiative MENA Grant Number: 100119.

**Competing interests:** No competing interest.

difficulties integrating the platform into teachers' schedules. The platform's animations were less effective in sustaining children's attention amid evolving digital standards.

## Conclusion

To enhance the platform's effectiveness, recommendations include simplifying the registration process, enhancing content interactivity, aligning the platform with school curricula, and equipping teachers with supportive resources. Fostering stronger school-family partnerships and engaging parents through community initiatives may be considered to maximize the platform's potential to promote healthier eating habits and improve nutritional awareness among children and their families, across the region.

---

## Introduction

In today's world, technology is increasingly prevalent, and children. are exposed to digital technology from an early age, with social media being integrated into their lives. Subsequently, there is a growing concern about the impact of media on children, as reported by teachers, parents, and health professionals [1]. Prior to the pandemic, digital interventions were already appealing to young people [2]. However, during COVID-19, the shift to online platforms intensified. Social media usage surged, particularly among youth, as individuals adapted to online learning and engaged more with health-related content. This period saw an expansion of digital health interventions and a growing reliance on social media for health information [3], making digital technologies even more prevalent.

Studies indicate that children and youth often prefer digital methods for learning about nutrition, viewing them as more relevant and impactful compared to traditional approaches, mainly due to their exposure to technology and the digital era [2]. Several platforms are available, including the internet, telehealth, gaming, social media, mobile apps, and wearable devices. Each of these technologies can play a pivotal role in enhancing children's health outcomes [2] and research from different regions of the world and age groups has shown promising results from digital educational interventions.

Research from different regions and age groups shows promising results from digital educational interventions. In fact, eleven website interventions targeting diet and physical activity (PA), providing nutrition education reported significant improvements in both diet and PA [4]. The KickinNutrition.TV (KNTV) program, an evidence-based digital nutrition and wellness curriculum designed for school-age children, has been shown to improve dietary behaviors and the ability to identify healthy dietary options [5]. This innovative curriculum utilizes digital technology, peer education, and online activities to enhance learning both at home and in school for middle school students. In Greece, the online educational initiative "Nutritional Adventures" has demonstrated effectiveness in increasing adherence to the Mediterranean diet, enhancing fruit and vegetable intake, and promoting PA among children [6].

A systematic review of six articles from different countries examined schoolchildren aged 8–16 and various types of games, including computer based, video games and

mobile applications. In three studies, children in the intervention group consumed and selected more fruits and vegetables (F&V). Similarly, in two other studies, the intervention group chose and consumed more healthy snacks compared to the control group. One study reported a decrease in sugar intake, although no significant difference was noted between the intervention and control groups [7]. These findings align with those discussed in another systematic review of fifteen studies on website interventions, which highlighted that the success of digital interventions for health improvement depends on multiple factors such as education, goal setting, self-monitoring, and parental involvement [8]. Adding to this, a meta-analysis of twenty randomized controlled trials, involving studies from the US, the United Kingdom (UK), New Zealand (NZ), Canada, and several others, further explored the impact of serious games on children's health. It examined the effect on body composition, physical activity, and dietary change in children and adolescents. In particular, 10 studies focused on physical activity, involving 489 participants in the intervention group and 425 in the control group, with intervention durations ranging from 1 to 7 months. Most of these studies reported a significant increase in physical activity, highlighting the positive impact of serious games. The dietary change studies, involving 1796 participants in the intervention group and 1913 in the control group, measured snack consumption, F&V intake, and sugar intake. Results indicated a significant reduction in snack and sugar intake, with a statistically significant increase in children's F&V consumption observed in only two studies [9].

The Ajyal Salima initiative is a school-based nutrition education program in the Middle East and North Africa (MENA) region, designed to raise awareness about healthy nutrition and physical activity in a fun and interactive way. It has been implemented in four Middle Eastern countries over several consecutive years, with recent findings published in a regional paper summarizing its efficacy on different outcomes across these countries [10]. Briefly, the Ajyal Salima program has proven successful in instilling concepts of healthy eating and physical activity in children 9–11 years of age and in changing food consumption behaviors and self-efficacy in this age group [10].

During the COVID 19 lockdowns, the need for a digital platform became even more evident, adding more value to the overall initiative and to the future updates of the Ajyal Salima program. The digital platform was developed to extend the program's reach beyond the traditional school setting, enabling students to access educational materials in the form of modules from home or other locations. The implementation of the educational modules is planned for the academic year, spanning around 3–5 months.

The primary objective of this pilot multi-center study was to qualitatively evaluate the usability and perceived value of the platform and identify potential gaps by gathering feedback from students, teachers, and parents, ultimately aiming to optimize its use as a tool to increase nutritional knowledge and improve children's dietary and Physical activity behaviors. Given the importance of parental involvement in the success of digital education, the Ajyal Salima platform targeted parents by inviting them to take part in their child's learning.

## Materials and methods

### Participants

The study included schools from four Arab countries (Lebanon, Jordan, Palestine and Bahrain) purposively selected to capture diverse perspectives on the implementation and impact of the Ajyal Salima program across multiple sociocultural and educational contexts. The schools were selected and contacted through the Ministry of Education within their respective jurisdictions, which was responsible for approaching schools, obtaining consent, and implementing the program locally. Schools were enrolled in the program based on their ability to conduct the intervention in accordance with the study protocol, the availability of necessary staff and facilities, including internet access and accessibility to connected devices at students' homes. Participants were school children aged between 9 and 11 years and enrolled in grades 4 and 5, along with their parents, classroom teachers and Ajyal Salima staff. The staff included field supervisors from participating countries responsible for conducting regular schools' visits to follow-up with teachers on the implementation of the

intervention. All staff members received research ethics training. Consent forms providing comprehensive information on the study's objectives, method, and risks were sent to parents through the schools. Parents signed consent forms for both themselves and their children. Only children with parental consent were approached for participation and all students were required to sign assent forms. Ethical approval of the study was granted by the Institutional Review Board of the American University of Beirut (AUB) in Lebanon (SBS-2021–0423).

Focus group discussions (FGDs) were conducted separately with children and parents. Across the four participating countries, a total of 21 FGDs were conducted with children (n = 145), including seven FGDs in Lebanon, five in Bahrain, five in Palestine, and four in Jordan. In parallel, 16 FGDs were conducted separately with parents, involving a total of 98 participants, including three FGDs in Lebanon, four in Bahrain, five in Palestine, and four in Jordan. Children participating in the FGDs were grouped by grade level (Grades 4 and 5) to ensure age-appropriate discussions, and participants were not mixed across schools. Each interview and focus group discussion lasted approximately 40 minutes, and thematic patterns were broadly consistent across grades. Parents and children were encouraged to share their opinions freely, and all participants were assured that all personal identifying information would remain confidential.

In addition, 18 in-depth interviews (IDIs) were conducted with teachers across the four countries (total interviewees: n = 19), including seven teachers in Lebanon, four in Bahrain, four in Palestine, and four in Jordan. Furthermore, eight key informant interviews (KIIs) were conducted with Ajyal Salima program staff, including three participants from Lebanon, one from Bahrain, two from Palestine, and two from Jordan. Table 1 details the number and distribution of participants, focus groups, interviews and schools across countries.

## Design and procedures

This qualitative feasibility study was run from May 2023 to May 2024 using focus groups and individual interviews. Participants recruitment took place in May 2023 in all participating countries simultaneously. The intervention was implemented over the course of one month, followed by data collection.

Teachers were introduced to the program and the digital platform through training workshops provided by the research team. While the platform was designed to be used at home, the teachers were instructed to assist children in setting up their accounts and logging in and monitoring their progress throughout the intervention period. Along with the consent letter, parents received a brief overview about the platform including its aim, content and instructions on how to register and log in their children.

## The intervention

The Ajyal Salima digital platform consists of a series of educational modules adapted from the existing offline classroom version [11]. The program focused on promoting healthy eating and an active lifestyle. Its focus areas included increasing

**Table 1. Total number of FGDs, IDIs and KII conducted and participants reached across countries.**

|  | Lebanon | Bahrain | Palestine | Jordan | Total |
|---|---|---|---|---|---|
| **Number of Schools** | 4 | 3 | 3 | 3 | **13** |
| **Number of students' FGDs** | 7 | 5 | 5 | 4 | **21** |
| **Total Number of students** | 43 | 28 | 37 | 37 | **145** |
| **Number of parents' FGDs** | 3 | 4 | 5 | 4 | **16** |
| **Total Number of Parents** | 12 | 20 | 37 | 30 | **98** |
| **Number of teachers' IDIs** | 7 | 4 | 3 | 4 | **18** |
| **Total Number of Teachers** | 7 | 4 | 4 | 4 | **19** |
| **Number of KII with Ajyal Salima Staff** | 3 | 1 | 2 | 2 | **8** |
| **Total Number of participants from Ajyal Salima Staff** | 3 | 1 | 2 | 2 | **8** |

fruit and vegetable consumption, having breakfast daily, minimizing the intake of energy-dense foods and beverages and engaging in regular physical activity. The intervention was based on the constructs of the social cognitive theory [12] which uses a multilevel approach involving individual changes and environment modification to support positive behavioral changes. Accordingly, the intervention comprised three components: (1) the classroom component including twelve culturally appropriate classroom interactive sessions addressing nutrition and lifestyle behaviors, (2) the family component consisting of meetings, health fairs and information packets helping families create a supportive environment at home and encourage healthy lifestyle behaviors and (3) the food service component targeting school shops and home prepared lunchboxes. The implementation of the three components in a coordinated way ensured that the intervention simultaneously targeted knowledge and self-efficacy at the individual level, role modelling at the family level and healthy food access in the school environment (Fig 1).

The online intervention, consisted of 10 learning modules which covered the following subjects: 1) Food groups and their benefits, 2) Food portion size, 3) Importance of fruits & vegetables, 4) How to stay physically active, 5) The importance of daily breakfast intake, 6) Tips for healthy snacks preparation, 7) Why water is the best beverage 8) Healthy food choices for dental health, 9) How to identify food rich in added sugars and fats 10) Learn how to choose Nutrient dense. Each module included animated videos and interactive games displayed on the Ajyal Salima Digital platform microsite. The animated videos include two main characters, a snail and a rabbit, who guide students through lessons in a fun and interactive dialogue format. After completing each module, children play a game to apply their newly acquired knowledge.

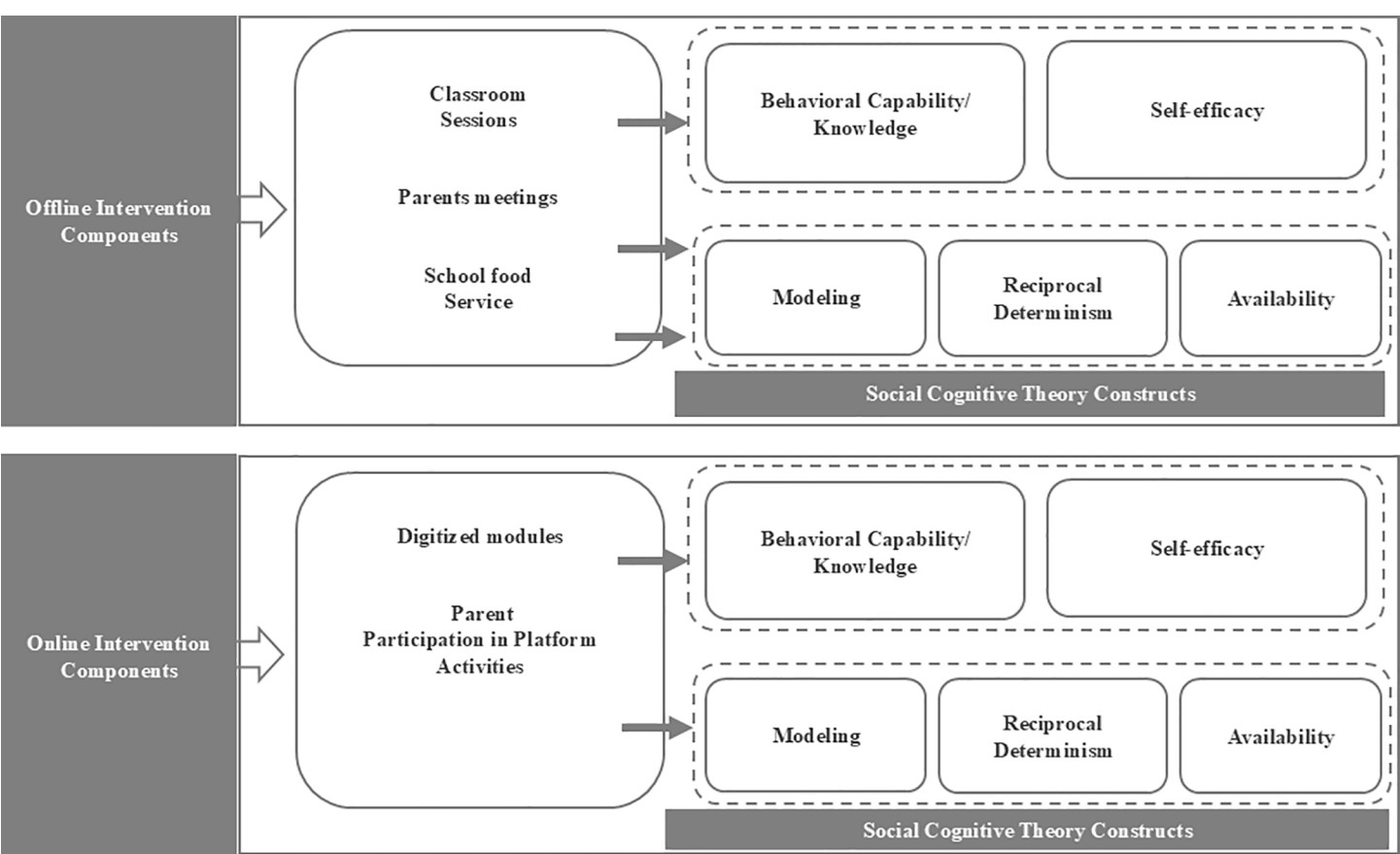

**Fig 1. Mapping offline and digital Ajyal Salima intervention components to Social cognitive theory constructs.**

The combination of animation and gamification was designed to make the learning more enjoyable, capturing students' attention and encouraging them to return to the platform for continued participation. Table 2 presents a Summary of Offline and Online Intervention Components.

The digital Ajyal Salima intervention was implemented during the academic year, following completion of school registrations and consent procedures. The platform is web-based and accessible via computers, tablets, and smartphones, allowing use both within and beyond the school setting. The intervention consisted of a sequenced set of interactive modules, each designed to be completed in approximately 15–30 minutes and delivered over a total intervention period of approximately 4 weeks. Modules were completed in a predefined order to support progressive learning, combining educational content, interactive activities, and reinforcement exercises. Teachers facilitated initial access to the platform, supervised classroom use when applicable and provided technical and motivational support to students throughout implementation. Parents assisted with technical access when needed; they were encouraged to support children's engagement at home by supervising platform use, engaging in certain activities and reinforcing key messages related to healthy eating and physical activity.

## Data collection

After receiving their consent and assent forms, students were provided with a code to access the online platform. Each code was unique to the teacher and her class, allowing the teacher to monitor and follow up on the progress of the students throughout the modules.

Students accessed the e-educational activities at home at their leisure; all the modules were completed during the intervention period, around one month. Although most students completed the whole program within a week, FGDs and IDIs were conducted with all participants one month following the completion of the intervention. Interview guides and procedures were consistent across countries. Data collection stopped when data saturation was reached.

Focus groups and interviews were conducted by independent interviewers appointed by each country's health or education ministry. These interviewers attended three training sessions, (1) One on Research Ethics, (2) a tutorial on how to use the Ajyal Salima microsite and (3) on how to conduct qualitative research.

Regular online meetings were held to collect and synthesize feedback from Ajyal Salima staff across different countries during and after the program's implementation. A regional debriefing session took place in Istanbul, Turkey, where team members from Lebanon, Jordan, Palestine, and Bahrain gathered to assess the platform's functionality, discuss implementation challenges, and identify potential solutions to improve user experience. This collaborative meeting enabled staff members to exchange insights and share best practices from their respective countries, fostering collaboration and making improvements for future developments.

## Data transcription

Focus group discussions and face-to-face interviews were conducted using a set of pre-determined core questions, however they were intentionally designed to encourage open (informal) conversations, allowing participants to express their views freely. Focus groups and interviews were conducted in colloquial Arabic to ensure cultural relevance and clarity for the participants.

To ensure accurate data collection, all interviews and focus group discussions were audio-recorded using digital voice recorders, with participants' consent. Recordings were transcribed verbatim in Arabic by trained, country-based

**Table 2. Summary of offline and online intervention components.**

|  | Target Population | Setting | Key Activities | Delivery Agent | Frequency |
|---|---|---|---|---|---|
| **Ajyal Salima Offline Intervention** | Children/Parents/Teachers | School | Material, Pamphlets, classroom activities | Teachers | Once per week |
| **Ajyal Saima Online Intervention** | Children/Parents/Teachers | Home | Online videos and animated activities | Autonomous use by students | Self-paced |

researchers. The anonymized transcripts were then securely shared with the research team at the American University of Beirut (AUB) for analysis. Transcripts were translated from Arabic into English by bilingual members of the research team.

## Data storage and confidentiality

The recordings made during the focus groups did not include names or identifiable information; they only included the questions and answers to the pre-determined interview questions and were not shared with any other party. All soft data were stored on a password-protected laptop, ensuring that only the researchers involved in the study had access to the password and, consequently, the data. As for hard copies/papers (used for documentation of focus group discussion, in Lebanon), they were securely stored in a key-locked room at the department of Nutrition and Food Science and only researchers included in this study had access to the papers. This approach ensured that all data was handled in a way that maintained participant privacy and adhered to ethical standards for confidentiality and data security.

## Data analysis

All transcribed data were analyzed using thematic content analysis, a method described by (Burnard, 1991). Thematic content analysis is adapted from grounded theory and was carried out using a systematic approach of immersion in data, coding, and data reduction. Quotes were inductively organized around four main themes: 1) *Usability and Support using the Digital Platform*; 2) *Content Enjoyment of Story Lines and Games*; 3) *Changes in Children's habits*; 4) *Recommendations to Improve the Digital Platform*; the mind map in the S1 Fig 2: Mind mapping of identified themes and sub-themes, illustrates the main themes and respective subthemes. NVivo software and established qualitative analysis procedures were used for data review.

## Results

Themes and their respective sub-themes were extracted, and quotes were organized accordingly. Major quotes with their identified source by country are presented in table 3 (S2 Table). Selected quotes were chosen to support the explanation of the results.

### Theme 1: Usability and support using the digital platform

**Sub-theme 1.1: Technical challenges and registration issues.** Most parents believed that the digital platform was easy to use with clear instructions, that their children needed a bit more time to adjust to the digital platform, but that the overall experience was positive and different from traditional ways of learning. However, some technical problems were encountered such as inserting email addresses, completing the registration process, accessing the link to the nutritional program as reported by many parents, children, and teachers. Collected quotes illustrate this observation, for instance, a parent in Palestine said that *"At first, we faced a problem with the email address where it would get mixed up when we wrote it down, but my husband fixed it for us"* and another in Jordan *"The account login process should be a bit easier than this, not too many accounts and complicated steps.".* This quote from a student in Lebanon supports this issue *"The password was not working".*

A recurring complaint was the glitching of the platform which prevented children from accessing it, interrupting their activities, or even requiring them to repeat activities. This was particularly noticeable in Lebanon, Palestine and Jordan, and was linked to factors such as unstable internet connections, the use of different devices or simply platform related problems. Some examples of children's quotes are listed below:

> *"I did not receive the certificate since the seventh activity glitched and I couldn't proceed. I replayed all the activities to make sure they are all unlocked, and this activity still did not unlock." (Student, Palestine)*

> *"I also have a slow internet connection which makes the platform glitch." (Student, Lebanon)*

These concerns were confirmed during teachers' IDI across all countries, particularly in relation to technical issues with the digital platform. They reported that many students had difficulties with registration, and needed support to sign in, create passwords, and/or access lesson plans or interactive games. For example, according to one teacher in Jordan, *"The issue was mainly with the registration process from the start. One of the problems was the nickname field".* Another teacher from Palestine shows the impact of these difficulties on the actual participation to the program by saying *"The students and their parents also need help since the registration process was a bit of a hassle".*

**Sub-theme 1.2: Parental and teacher support – Role of teachers and parents in helping children adapt to the platform.** Despite the reported issues in registration and some activities, all parents believed that their children received continuous technical support when needed, especially by their teachers, who were, according to parents, always available to help and maintained regular communication regarding the digital platform. This was particularly obvious in this quote from Bahrain: *"Yes, there was great support from the teachers, as they even provided us with instructions on how to log into the platform. They also maintained continuous communication with us through contact channels".* Parents of other countries also mentioned some support from other family members although most of the support came from teachers. In Palestine for instance, a parent mentioned *"My daughter had some help from her aunt but was able to proceed on her own later"* However, the teachers complained that the registration process and follow-up was overwhelming given their overloaded schedules *"Time is a conflict since there is already too much to teach and little time to do so."*

While the registration phase posed some challenges, many students were able to navigate the digital platform independently once registered, as this Jordanian student stated *"Usually, my mom opens the platform for me and I play on my own".* However, most students mentioned requiring repeated assistance for logging in, or during some activities, be it for content or for technical issues such as glitching. Students generally thought that teachers and parents cooperated to provide support when needed. Many teachers provided follow-up discussions around healthy eating and activities.

## Theme 2. Content enjoyment: Storylines and games

**Sub-theme 2.1: Engagement with games, characters and design preferences.** Overall, students liked the platform's digital format, which included interactive games, videos, pictures, and the characters. They found the platform both entertaining and educational, and several students wanted to involve other friends or siblings to join. This is represented in the following quotes from students in Palestine *"I would recommend it to younger age groups because it is fun and so that they stay away from unhealthy foods. The platform itself is nice and encourages us to play".*, Jordan *"I would play again but it would be better if you updated it and added more videos and new information that we can benefit from. My younger siblings used to watch some of the easy videos with me so they can learn from them too",* and Lebanon *"Their voices were funny, and they explain very well. The information was easy."*

All parents believed that the content was age-appropriate and that the games and videos on the platform were new and attractive to their children. One parent in Palestine thought that *"Everything was suitable for their age group and the characters were nice and entertaining".* In Jordan, parents were also very positive about the appropriateness of the format and content of the platform *"The information is presented in a simple and nice way using cartoon characters which is nice and suitable for kids"*, *"The platform is using an approach similar to video games which is attractive to children and makes the information stick to their minds."*

Some parents liked the point collection system, believing it encouraged children to participate and stay motivated. One parent from Palestine said, *"The concept of collecting points was also a great idea that created a competition between the girls."* Another parent from Jordan specified *"I think what attracted the children most to the platform are the things that can include competitions like the games"* and recommended by some parents as further improvement. Notably, in Bahrain one parent said, *"The program should be continuous, with different levels, and the addition of some competition activities".*

Parents also expressed that the use of vibrant colors and appealing characters as well as the varying difficulty levels in online games contributed to their children's motivation. One parent from Bahrain said *"The characters were fun and made*

learning exciting, our favorite character was Fareed the rabbit. The information was easy to understand, and some of it was already covered in the family education subject". According to teachers, in the beginning, some students were not particularly excited about the digital platform but became more motivated once they started the games. For example, one teacher in Bahrain said "It was a new experience, but the students enjoyed it and were excited to complete the activities. It was a good alternative to electronic games." The cartoon characters and interactive games did catch their attention: "They enjoyed it because of the existing drawings, cartoons and videos" as per a teacher from Jordan.

**Sub-theme 2.2: Challenges with content – Difficulty with certain lessons and lack of engagement in certain videos.** Overall, this theme highlighted differences between countries, with Palestine reporting the most hardship with some activities. Also, the age of children plays a role in the appreciation of videos and the appropriateness of the digital interface.

Although the content of the platform was considered easy as seen in previous sections, some challenges were reported in the different focus groups. The difficulties mentioned by parents were generally subject-specific. For instance, in Palestine and Bahrain, the food portion activity was deemed difficult as evident from the following quotes "She had some difficulties with the food portions activity" or "The most challenging games were "food portions," and the "maze dice". A common difficulty mentioned by students in Palestine was the lesson on fats, which some found challenging due to the speed at which the information was presented as mentioned in the following quotes "The video where they were talking about the fats and oils was a bit hard.", "All the stories were easy except for the one about fats." This was not the case in Lebanon and Jordan where children thought the activities were easy, more suitable for younger kids, and sometimes boring. A student in Jordan said: "I suggest making the activities a bit harder because I thought they were too easy and students younger than us could do them". Similarly, a student in Lebanon stated, "I felt it was for younger kids". Finally, some students, generally older, in grade 5 lost interest in the digital interface and found the interactive design less engaging. Some students reported that videos lasting between 4–6 minutes or longer, did not hold their attention, leading to disengagement, one student in Lebanon said, "One of the videos was boring I had to leave the phone and come back to do the activity".

One teacher in Lebanon confirmed the lack of motivation of students to use the platform by highlighting the importance of finding different ways of engagement and motivation.

**Sub-theme 2.3: Integration with classroom learning – Teachers' perspectives on how the platform complements traditional teaching.** The use of this digital platform in the classroom is viewed to be a valuable complement to the curriculum, a means to break away from the rigidity of traditional teaching and introduce more interactive and flexible learning experiences. Many teachers believed that the content of the platform could be easily integrated into the classroom, more specifically in science class. As one teacher from Palestine explained, "As a science teacher, I can include the food groups in my classes by using the lessons, videos and games from the platform, in case a computer was available." Another teacher from Jordan "Sure, for the future, I can benefit from the platform in my lessons. I tackle food groups in my classes, like carbohydrates and proteins. As for the fourth grade, there are full lessons about these subjects". In Lebanon, teachers also had a positive attitude towards using the platform as a teaching tool "Sure especially when explaining the food pyramid, I could use some of the games as a supplemental teaching tool." Another teacher in Lebanon valued the practical aspect of the platform compared to conventional learning saying that "I can benefit a lot from this platform in the classrooms. It's much easier than how we used to do it before—printing posters, bringing materials, and you know, with the current economic crisis, this feels like a more practical solution". Interestingly, in Bahrain, teachers were more proactive having already integrated the program in their curriculum or side projects. For example, one teacher mentioned "I integrated the Ajyal Salima curriculum into the Healthy Mind in a Healthy Body project, as well as into the Family Curriculum".

Although parents did not express their opinions about this topic in most countries, quotes collected in Jordan highlight the parental support for this platform to be used in the classroom. For instance, one parent suggested "… implementing the platform's activities in class too by including one session per week dedicated to this."

 

### Theme 3: Changes in children's habits

**Sub-theme 3.1: Nutritional behavior changes and knowledge-action gap.** Parents believed that their children were choosing healthy foods more often and avoiding take-away meals and processed foods. For instance, a parent from Palestine said *"She stopped buying from the school's cafeteria and started making herself salads and eating more fruits and vegetables. She also bought less foods from the extra group."* And another stated *"She started eating more fruits and vegetables, cut down on buying junk food, started eating more sandwiches and yogurt."* Children also showed an increased interest in food preparation or being present in the kitchen during meal cooking. In Jordan for example, one parent reported *"She's making her own sandwich that included healthy foods like labneh and tomatoes"* while a parent from Palestine said, *"She started making her own lunchbox and got used to eating breakfast, which inspired her sister to do the same."* The influence of children's growing knowledge on the rest of the family transpired in Palestine and Jordan with one example of parent from Palestine *"... she made all of us cut down on unhealthy foods".* Similarly, in Jordan, one parent said *"My daughter started forcing us to eat a side salad every day. She began to cut the vegetables herself saying that we need to include salads in our lunch meal. She also started to cut herself some fruits saying that this is healthy instead of eating dinner."*

All teachers believed that Ajyal Salima platform contributed to improving nutritional knowledge which reflected positively on students' dietary habits; this included healthy food preparation and healthier alternatives in lieu of fast food and sugary drinks. For example, according to one of the teachers in Palestine, *"I noticed an increase in their consumption of healthy foods like fruits and vegetables since they focused on getting 5 servings per day".* Similarly in Bahrain, a teacher stated *"Students' nutritional habits and concepts have changed. They became more consistent with having breakfast, avoided fast food, soft drinks, and consuming from the extras groups…."* In Jordan, teachers reported similar positive behavioral changes with children.

Students reported noticeable changes in their dietary habits, including reduced sugar intake, fewer snacks, better portion control and increased physical activity. For example, one child from Palestine said *"I started drinking more water and eating more fruit and vegetables. For instance, I ate 4 carrots yesterday because it was mentioned on the platform"* This was also reported by a student from Lebanon *"I used to eat a lot of sweets, but after watching the video and doing the activities, I learned a lot."*

Student from Bahrain specified *"...We committed to having breakfast, eating more fruits and vegetables, avoiding soft drinks, and playing sports."*

Finally, teachers thought that students who completed the program seemed to have more knowledge and awareness around healthy eating and influenced their peers in making healthy choices. A teacher in Palestine explained *"Their knowledge increased, and we did a revision of the food groups. As for their food intake, they cut down a bit on the foods belonging to the extra food group".*

Despite the positive feedbacks from parents and teachers, some statements collected in focus groups disagreed with the above. Indeed, a few teachers stated that while knowledge and awareness around healthy eating increased, actual implementation did not. This was recorded in Jordan where two different teachers emphasized the contrast between the knowledge and its application in the following quotes: According to two teachers in Jordan, no major differences were noticed in children *"No, they are still buying chips and chocolate. They are aware and understand, but in terms of application they did not".* Similarly, in Palestine, a couple of parents reported not seeing any changes in their children's habits, despite better knowledge. For example, one parent said *"We are still learning... but the implementation is slow... She cut down on drinking sodas to only once per week."*

**Sub-theme 3.2: Physical activity changes.** Parents agreed that their kids' physical activity levels improved after using the digital platform. In fact, many statements reported that children started exercising more often, playing sports daily, jumping rope, using a bicycle, or following the sports video routines. Parents reported increased energy, daily movement,

and enthusiasm for sports. For instance, in Palestine, one parent stated, *"She wants to play more sports willingly, which felt like an obligation for her before using the platform"*. In Jordan, when asked if they had noticed any changes, one parent in Jordan said *"Ah, a lot, a lot, especially sports. From the day she saw the sports (activity), she got excited."*

Children mentioned explicitly that they started exercising more and even at specific times like morning routines. A student from Jordan explained *"I started doing more physical activity since I used to be more sedentary. Watching the videos motivated me to be active and taught me that sports are extremely beneficial to our body."* Similarly in Palestine, a child reported an increase in physical activity *"I started playing more sports"*. In Bahrain and Lebanon, although the collected quotes were limited, they were aligned with those collected in Jordan and Palestine.

The sports activity was for some students an opportunity to share a moment of play with siblings or friends, for example one parent from Palestine said that her daughter *"Started doing Zumba dances with her sister...* and another in Jordan said *"... She also started moving more instead of sitting all the time."* This was confirmed by a couple of student quotes from the same countries.

### Theme 4: Recommendations to improve the digital platform

This theme provided valuable insights, offering critical feedback for the research team on areas for improving the digital platform. The recommendation themes listed below cover issues beyond technical improvements where some recommendations were already mentioned in theme 1.

### Sub-theme 4.1: Enhancing platform features and implementation

Parents provided positive feedback on the digital platform and also suggested some recommendations to improve its effectiveness. These included expanding accessibility of the platform to more age groups as a parent from Palestine said, *"I suggest developing the platform to make it accessible for all age groups"* and a parents from Jordan *"I suggest introducing the platform to even younger age groups so they can learn at a younger age how to lead a healthy lifestyle"*

Parents recommended to include sharing features allowing children to share photos, videos or progress on activities but also the application of their learnings at home. Some parents even suggested setting up a WhatsApp group for parents to support their children's progress. For example, one parent from Jordan suggested *"...I suggest also creating a WhatsApp group at the beginning of the school year that includes some mothers who can supervise the progress on the platform."* In Palestine, one parent said, *"I suggest adding an option where students can upload pictures and videos of the activities they do at home"*. To which three other parents agreed. Parents also concurred that gamification and competition could further motivate their children. By adding points, rewards, and rankings, parents believed that their children's motivation would be increased. For example, a parent in Bahrain stated, *"The platform needs challenge-based activities between two people to increase excitement, along with (notes) to allow self-reflection."* Although facilitating the registration process was quite a frequent suggestion, the most common recommendation from parents was to improve and customize characters and make the platform accessible for younger age groups, the quotes from which have been covered above. This view was reinforced by students: *"I wish they added one more character to the videos"* from a child in Palestine, and *"...We can choose the characters ourselves and customize their clothing, name and even characters where we can choose the animals we like"* from a child in Jordan.

As for teachers, they proposed some ways of enhancing the learning potential of the platform such as this teacher in Palestine who said *"I wish there was a practical activity in between games because I felt that the overall picture wasn't clear"* or a parent from Jordan who suggested *"…. Another thing that can be added is a section at the end in which the child has to do a project or maybe draw a picture that depicts how they benefited from the platform."*

Students shared a range of ideas focused mainly on improving engagement, content variety, and making the platform more fun and personalized. One meaningful suggestion that aligns with recommendations made by parents is to add more

 

games, levels, rewards, and interactive features. In Bahrain one child said, *"We also need more motivation, additional levels, more points, and increased opportunities for two-player challenges."* And in Palestine, *"I suggest adding extra levels".*

Improvement of cartoon characters was a recurring theme for parents, teachers and children. All participants suggested introducing new characters as students' progress through the platform to maintain interest and introduce variety. In addition, some suggestions were made to customize characters through different clothing, names etc… For example, a parent in Palestine said that *"I suggest adding more colors and characters".* In Bahrain, a teacher suggested *"adding more characters to engage children further."* And in Lebanon, one teacher's opinion about the interface was that it needed to be enhanced on the digital side to remain interesting for the children, *"In my opinion the videos and activities are neutral since there are much more sophisticated and complicated ideas online".* Some suggestions were also put forth by parents to develop character-driven stories, turning each learning unit into an episodic narrative to make the platform feel like an ongoing journey. A notable quote from a parent in Palestine *"... It's like creating a 10 episodes series for example with a united background story by changing the way the story is told".* Finally, one suggestion from a parent in Palestine was to make characters more relatable by changing to a real-life character format. Children also agreed on this, a child from Bahrain said, *"we need more engaging characters and additional levels".* Some children also expressed their wish to be introduced to more recipes *"I suggest adding a section in the platform where we can learn how to make healthy versions of foods, like chips for example.*

Finally, a common recommendation, especially amongst teachers, is to change the timing of implementation, deemed inappropriate as it is too late in the year. In Lebanon, one teacher recommended to start the program earlier, as mentioned in this quote: *"... by the end of the school year, students tend to lose motivation and energy."* Teachers from Palestine and Bahrain confirmed that the timing of implementation was not ideal. In Palestine, one teacher said *"The implementation time happened to be during the exams period and the beginning of Ramadan so there wasn't a lot of time...".*

The length of implementation was also a subject of a couple of recommendations from teachers but also parents and students, wishing the program was implemented during the whole school year as some students finished all the activities within a week or less. Both parents and teachers believed that students would have benefited more if the period of implementation had been longer.

### Sub-theme 4.2: Parental involvement – Challenges and recommendations

Although parental involvement did not emerge as a meaningful theme in the recommendation theme, discussions in focus groups highlighted challenges and barriers in engaging parents to support their children during the implementation of the platform. To these challenges, some recommendations were offered by parents and teachers. In Jordan for instance, one parent stated, *"The platform is nice but needs follow up and commitment from parents",* while a teacher suggested to raise parental awareness *"Parents must know whether it is good or bad to pack a chocolate sandwich in their kid's lunchbox."* Another teacher in Lebanon went further to suggest how to raise awareness of the platform *"An open house for parents (for example for an hour) to introduce them to the platform and show them what the program is about".*

Parents were generally not very involved or engaged in their children's use of the platform. Many of them admitted to either not using the platform themselves or not following up on their children's activities. For example, a parent from Lebanon, said *"No, I didn't notice the platform, but my son went in and tried, and he started making food...".* The teachers noted this parental absence; a teacher in Jordan said *"They weren't much interested in it. Maybe because it is not part of the curriculum or because it happened to be implemented at the end of the school year."* Similarly, in Lebanon, teachers reported that parents were not following up with their children's progress. This is reflected in the following quote

*"The parents were not involved in the platform and did not follow up on the activities." and "I was shocked that the parents did not follow-up with their children and did not even check what they are doing on the platform".*

Although not many children quotes were collected, they confirmed the low engagement of their parents and their lack of support. More importantly, this lack of support was sometimes a barrier to their participation in the program. this was particularly noticeable in the following respective quotes from two students in Jordan: *"I did not do the shoebox activity since my mother was busy and I couldn't do it on my own"* and *"I couldn't try any recipe because my mother doesn't allow me to access the kitchen and did not let me try any recipe".* In Lebanon and Palestine, similar barriers were encountered as seen in these respective quotes from children *"Every time I ask my mom something, she tells me, later, I'm busy right now"* and *"There was a lot of information given at once, and no one helped me at home".*

**Feedback collected from Ajyal Salima Staff.** Insights from Ajyal Salima Staff meetings highlighted the various challenges encountered during the implementation of the digital platform. First, the registration process posed the greatest obstacle to accessing the platform. This issue was compounded by various technical difficulties encountered by students, parents, and teachers, limiting the number of participants and their level of engagement. Access to the internet for example was a crucial issue where some families were limited by internet access or internet usage in their homes, as reflected in this quote: *"Internet connection was always an issue."*

Initially, the platform aimed to target parents, but unfortunately, there was notably low participation from parents with the platform activities. This could be attributed to several factors, such as time constraints, competing activities with their children, or insufficient access to tablets or phones for their kids to use. This concern was clearly expressed by staff, for example: *"Some parents understood the overall project but did not follow their children progress or engage with them in the activities. They received information from their kids, saw them doing some activities, and that was it."* Low parental involvement can also be due to the fact that the program was being implemented in schools, which made the parents rely on the teachers and schools to constantly follow up on their children's progress. Many teachers suggested parent meetings as an important step to describe the objectives of the intervention and expectations around parents' involvement, as illustrated by the following quote: *"Parent meetings are crucial for explaining the type of intervention taking place, how they should be involved, and how to use the platform. Informing them through a letter or relying on kids to inform them has not been effective."*

Teachers complained about overwhelming schedules, which hindered them from keeping track of and constantly following up on children's progress. They also faced issues persuading kids to register, engage in activities, and learn through the platform. In concordance, convincing parents to register and monitor their children's performance and activities was challenging. One staff member summarized this constraint by stating: *"Some teachers faced issues with the program because they were overloaded and had to prioritize the curriculum over activity programs."*

Several other barriers hindered the proper engagement of young users. One significant challenge is the increasing difficulty of creating animations that meet children's evolving expectations and capture their attention. As children are increasingly exposed to sophisticated augmented reality games, simple or basic animations may fail to engage them effectively. Despite these challenges, staff acknowledged the added value of the platform once technical barriers were addressed, as highlighted in this quote: *"The platform is an additional asset to the program, once the registration issues are resolved."*

## Discussion

This qualitative study explored the feasibility and perceived usefulness of the digital Ajyal Salima program from the perspectives of children, parents, and teachers. Overall, findings indicate that the platform was viewed positively, particularly for younger children, who found the content engaging and age-appropriate. Parents and teachers generally perceived the platform as useful and easy to navigate, although technical difficulties related to registration, internet connectivity, and access to games emerged as major challenges. Children appreciated the playful and collaborative nature of the program, while older students reported lower engagement with the animations. Although behavior change was not a primary outcome, both children and parents perceived improvements in dietary habits and physical activity. Low parental engagement and increased reliance on teachers were identified as important barriers affecting usability and implementation.

The technical challenges identified highlight the critical role of usability and digital literacy in determining the success of digital educational interventions. Difficulties varied across countries, underscoring differences in access to devices, internet quality, and technological literacy, all of which are closely linked to socioeconomic status. These findings align with existing literature demonstrating that usability is strongly associated with engagement and improved outcomes in digital health and educational platforms [13–15] be it in the field of health behaviors, creativity, or skill acquisition. A review of 610 studies found that usability evaluation methods directly influence the success of digital health tools, with user-involvement and tailored approaches improving outcomes [15]. Usability is also closely tied to eHealth literacy, which influences users' ability to access, understand, and apply digital health information [16–18]. In the context of Ajyal Salima, limited access to connected devices, unstable internet connections, and varying levels of digital literacy directly affected feasibility, suggesting that these contextual factors must be considered when scaling or adapting the program.

Children's feedback emphasized the importance of age-appropriate design and sustained engagement. While younger children responded positively to the characters and animations, some older children found the content less appealing. This reflects the broader challenge of designing digital content that meets the evolving expectations of older children, who are increasingly exposed to sophisticated digital games and platforms [19,20]. These findings support recommendations in the literature to actively involve children in the design and development phase to ensure that content, character design, and video length align with users' preferences and developmental stages.

Although not designed to evaluate effectiveness, the study yielded insights into perceived behavioral changes. Both children and parents reported reductions in sugar intake, snack consumption, and portion sizes, alongside increased physical activity. These perceptions are consistent with evidence from school-based nutrition and physical activity programs, which have shown improvements in dietary behaviors and knowledge [21–23]. Similar outcomes were previously reported for the Ajyal Salima offline program using standardized measures [10]. While these findings should be interpreted cautiously, they suggest that the digital platform may serve as a promising vehicle for delivering nutrition and physical activity messages, and could be effective in implementing behavioral changes. The impact of the digital platform on eating behavior and physical activity should be measured in an intervention trial measuring the benefits perceived in this qualitative feasibility study.

Engagement is a fundamental dimension in behavioral programs and digital health apps for weight management, nutrition and physical activity promotion [24,25]. Sustained engagement, combined with features like interactive feedback and parental involvement, is also essential for efficacy. Sustained engagement of children is generally promoted by goal setting, self-monitoring within the duration of exposure, and parental involvement, contributing to obtaining the best results in terms of behavioral change [8]. The goal setting aspect of engagement was highlighted in our study, especially in quotes collected from parents saluting the point system and other features that fostered competition amongst peers. According to parents, these features should be further developed as they provided motivation to learn through the platform.

Parental involvement emerged as a key determinant of platform use and effectiveness. Despite activities designed to involve parents, participation was low, increasing reliance on teachers and contributing to perceived workload. The importance of parental engagement in child health interventions is well documented, with evidence showing positive associations with nutrition and physical activity outcomes across socioeconomic contexts [26–31].

The 2015 Global School Health Survey conducted in Oman, which focused on parental involvement and adolescent well-being showed a significant correlation between positive parental involvement and nutrition and exercise. Moreover, higher parental involvement was associated with higher odds of good nutrition. Larger studies from the United States and Eastern Mediterranean Region indicate that higher family functioning is linked to better nutrition habits [30]. Another study on Australian adolescents further supports these results, highlighting that maternal knowledge of healthy food items and their availability at home, is related with higher intake of fruits and vegetables and lower consumption of energy-dense snacks [30]. These findings suggest that increased parental involvement is associated with better physical health (i.e., improved nutrition and physical activity) [31].

The impact of parental engagement is far beyond the raising of the child's interest in the platform. It affects children learning and teacher involvement. This impact was actually reflected in the reduced usability of the platform. A lack of parental engagement and support at home with technical difficulties can lead to disengagement of children, consequently impacting nutrition and PA learning, knowledge, and behavior. Barriers to parental involvement identified in this and previous studies include time constraints, limited digital skills, cultural perceptions of schooling, and communication challenges between schools and families [32–35]. These barriers may be particularly relevant in digitally delivered programs, where parental support is often required to overcome technical challenges. Narrowing it down to the Arab world, seven barriers to parental involvement were identified: time restrictions, student attitudes, poor communication skills, student performance level, parents' characteristics, parents' attitudes, and unappealing activities [32].

A lack of parental support was also noted as a reason for the reliance on teachers' support and potentially causing the teachers' work overload mentioned by some during the focus groups and the staff feedback meetings. In fact, previous qualitative research exploring the use of digital platforms for behavioral education in children report that the success of these programs is determined by multiple important factors including a commitment from the school leadership to prioritize such programs and allow time and training for teachers and parents to be able to support the digital initiative [36,37]. This time allocation would allow teachers to properly integrate sessions dedicated to digital learning in the curriculum to follow up on the progress of children, solve issues they are facing, and motivate them to keep learning. This was a common suggestion made by teachers and parents in this study.

Given our observations of low parental engagement, reliance on teacher support and their perceived impact on the use and effectiveness of the platform, we believe that finding the right balance between parental and teacher support to students, may be one important improvement to bring to the program.

From a practical and policy perspective, these findings highlight several implications. First, digital school-based health programs should be adapted to local technological and socioeconomic contexts to ensure equitable access. Second, sustained engagement strategies, including content expansion and controlled pacing, should be integrated into platform design. Third, balancing parental and teacher support is essential to avoid overburdening teachers while ensuring children receive adequate guidance. Identifying effective ways to engage parents—through needs assessments, practical content, and flexible online meetings or webinars—may enhance program sustainability and impact [33,38,39]. Future research should include targeted studies on parental engagement strategies and intervention trials to assess the long-term effectiveness of the digital Ajyal Salima program on behavioral outcomes.

## Strengths and limitations

The implementation of the program during this phase demonstrated several strengths, notably the collaboration with local partners and the successful engagement of schools, which facilitated access to the digital platform. The platform's interactive format and educational content were generally well-received, especially by younger children, and supported the delivery of key nutrition and physical activity messages. Moreover, the platform proved to be a potential educational tool not only for children but also for teachers and parents, offering opportunities for reinforcing nutrition messages through engaging digital content. However, several limitations were identified, the most important ones being the length of the intervention and parental engagement channels. The duration of exposure to the platform represents an important limitation. Some children completed all modules within a short period of time (in some cases one week), limiting sustained engagement. Evidence from digital behavior change interventions suggests that consistent use over several months is often required to achieve meaningful behavioral change, particularly for nutrition and physical activity outcomes [8,40,41]. Another limitation was at the level of translation of transcripts from Arabic to English. There was no back translation to Arabic which may have introduced the interpreter's bias. However, the forward translation was reviewed by bilingual researchers to ensure cultural relevance of the translated content. Finally, the purposive sampling procedure may have introduced a selection bias, given that schools were selected based on their logistical capacity to participate in the study,

including internet access and students' access to connected devices. This may have influenced representativeness; however, such criteria were required to ensure meaningful engagement with the intervention. These criteria were essential to ensure feasibility, consistent delivery of the intervention and reliable data collection.

## Conclusion

In conclusion, the results of our study support the usefulness and usability of this digital platform for nutrition and physical activity education, especially that it fulfills the need for children to learn in a playful way. In this paper, we highlight the challenges and limitations of the intervention in special developing countries' settings and provide several next steps to improve its implementation or similar educational platforms. First, further refinement of the intervention should focus on better assessing and addressing the socioeconomic context and digital literacy of households in developing country settings, alongside improving platform usability and content adaptation to children's digital experience expectations. Second, feasibility and piloting efforts should examine strategies to strengthen school–family partnerships, simplify registration processes, and enhance children's engagement, while systematically assessing reach, acceptability, engagement metrics, and implementation fidelity. Third, given the interest expressed by teachers, the use of the platform as a classroom-based, hands-on learning tool warrants further evaluation through effectiveness studies, potentially using quasi-experimental or cluster-based designs to assess impacts on children's nutrition knowledge, attitudes, and behaviors. Finally, implementation and scale-up considerations should include extending the platform beyond the school setting through active parental involvement, community-wide campaigns, and collaborations with local governments and media, as well as ensuring teachers are provided with adequate time, training, and resources to support sustainable integration. Together, these steps may enhance the platform's potential to promote healthy eating and physical activity habits among children and their families.

## Supporting information

**S1 Fig. Fig 2. Mind mapping of identified themes and sub-themes.**
(TIF)

**S2 Table. Table 3. Quotes from parents, teachers, and students categorized by themes and sub-themes.**
(DOCX)

**S3 Checklist. COREQ checklist. Consolidated criteria for reporting qualitative research.**
(DOCX)

## Acknowledgments

We thank the Ministries of Education in Lebanon and Palestine, Ministries of Health and Education in Jordan and Bahrain; the schoolchildren, their parents and their teachers; The Royal Health Awareness Society in Jordan for data collection and securing approvals; Dina Mansour and Anna-Maria Dannaoui for referencing and proof reading. We also thank Sonia Najem, Hilda Khoury and Fadi Yarak from Lebanon; Reem Jarrar, Dr. Samar Batarseh and Khitam Hattar from Jordan; Nisreen Khaleel Rimawi, Furat Muntaser Ateeq, Lubna Daoud Awwad, Maysa Khalid Al Younis, Sojoud Fawzi Jabra, Sanaa Othman Qawasmi and Aisha Deeb Ghazawi from Palestine; Dr. Mariam Ebrahim AL Hajeri, Dr. Ashwaq Abdulla Sabt, Hessa Khalid ALshaik, Aysha Mohamed Salim, Feda Ahmed AbdulRasool, Lulwah AbdulAziz AlThakir, Saad Eid Saleh, Ahmed Saad Ali, Hameeda Foad AlMudaweb and Enas Salem AlKhalaqi from Bahrain.

The intervention was supported by the Nestlé for Healthier Kids initiative – Nestlé Middle East (Grant Number: Award 100 119). The latter had no role in the research design of the study; in the collection, analyses or interpretation of data; in the writing of the manuscript or in the decision to publish the results.

## Author contributions

**Conceptualization:** Carla Habib-Mourad, Carla Maliha, Nahla Hwalla.

**Data curation:** Eman Haji, Lina AlTarazi, Suzan Totah.

**Formal analysis:** Amira Kassis, Diala Tailfeathers.

**Funding acquisition:** Carla Habib-Mourad, Nahla Hwalla.

**Methodology:** Carla Habib-Mourad, Amira Kassis, Diala Tailfeathers, Marco Bardus.

**Project administration:** Carla Maliha.

**Software:** Diala Tailfeathers.

**Supervision:** Carla Habib-Mourad, Marco Bardus, Eman Haji, Lina AlTarazi, Suzan Totah.

**Validation:** Carla Habib-Mourad.

**Writing – original draft:** Carla Habib-Mourad, Carla Maliha, Amira Kassis, Diala Tailfeathers.

**Writing – review & editing:** Carla Habib-Mourad, Carla Maliha, Amira Kassis, Diala Tailfeathers, Marco Bardus, Eman Haji, Lina AlTarazi, Suzan Totah, Nahla Hwalla.

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
