## [Decision Letter · Decision Letter 0]

25 Dec 2025

Dear Dr. Habib-Mourad,

We look forward to receiving your revised manuscript.

Kind regards,

Nour Amin Elsahoryi, pHD

Academic Editor

PLOS One

Journal Requirements:

“The intervention was supported by the Nestlé for Healthier Kids initiative – Nestlé Middle East (Grant Number: Award 100 119). The latter had no role in the research design of the study; in the collection, analyses or interpretation of data; in the writing of the manuscript or in the decision to publish the results.”

“Nestle For Healthier Kids Initiative MENA Grant Number: 100119”

“Nestle For Healthier Kids Initiative MENA Grant Number: 100119”

6. Please include a caption for figure 1.

7. We note that this data set consists of interview transcripts. Can you please confirm that all participants gave consent for interview transcript to be published?

If they DID provide consent for these transcripts to be published, please also confirm that the transcripts do not contain any potentially identifying information (or let us know if the participants consented to having their personal details published and made publicly available). We consider the following details to be identifying information:

- Names, nicknames, and initials

- Age more specific than round numbers

- GPS coordinates, physical addresses, IP addresses, email addresses

- Information in small sample sizes (e.g. 40 students from X class in X year at X university)

- Specific dates (e.g. visit dates, interview dates)

- ID numbers

Or, if the participants DID NOT provide consent for these transcripts to be published:

- Provide a de-identified version of the data or excerpts of interview responses

- Provide information regarding how these transcripts can be accessed by researchers who meet the criteria for access to confidential data, including:

a) the grounds for restriction

b) the name of the ethics committee, Institutional Review Board, or third-party organization that is imposing sharing restrictions on the data

c) a non-author, institutional point of contact that is able to field data access queries, in the interest of maintaining long-term data accessibility.

d) Any relevant data set names, URLs, DOIs, etc. that an independent researcher would need in order to request your minimal data set.

For further information on sharing data that contains sensitive participant information, please see: https://journals.plos.org/plosone/s/data-availability#loc-human-research-participant-data-and-other-sensitive-data

If there are ethical, legal, or third-party restrictions upon your dataset, you must provide all of the following details (https://journals.plos.org/plosone/s/data-availability#loc-acceptable-data-access-restrictions):

1. A complete description of the dataset

2. The nature of the restrictions upon the data (ethical, legal, or owned by a third party) and the reasoning behind them

3. The full name of the body imposing the restrictions upon your dataset (ethics committee, institution, data access committee, etc)

4. If the data are owned by a third party, confirmation of whether the authors received any special privileges in accessing the data that other researchers would not have

5. Direct, non-author contact information (preferably email) for the body imposing the restrictions upon the data, to which data access requests can be sent

Reviewers' comments:

Reviewer's Responses to Questions

**Comments to the Author**

1. Is the manuscript technically sound, and do the data support the conclusions?

Reviewer #1: Yes

Reviewer #2: Partly

2. Has the statistical analysis been performed appropriately and rigorously?

Reviewer #1: N/A

Reviewer #2: N/A

3. Have the authors made all data underlying the findings in their manuscript fully available?

Reviewer #1: Yes

Reviewer #2: Yes

4. Is the manuscript presented in an intelligible fashion and written in standard English?

Reviewer #1: Yes

Reviewer #2: Yes

Reviewer #1: Overall, this manuscript addresses a valuable and timely topic with promising potential for digital nutrition education. With focused revisions to clarify methodology, streamline results, and deepen interpretation, the manuscript will significantly strengthen its contribution to the field. The manuscript needs major revision mainly to improve the methods, result and discussion sections.

Reviewer #2: Thank you for the opportunity to review the manuscript; “Successes and Challenges of an online based nutrition awareness program in 9–11-year-old children In Four Arab Countries: The Ajyal Salima digital platform Qualitative study”. The paper has the potential to add detailed insights for adapting and implementing a digital version of the Ajyal Salima program. Below are my comments to help strengthen clarity, completeness of reporting, and interpretability of findings.

Major comment

1. Use a qualitative reporting guideline (e.g., COREQ [1]).

- Please select and apply an established reporting guideline for qualitative studies. The COREQ (Consolidated Criteria for Reporting Qualitative Research) checklist would be appropriate for interview and focus group–based qualitative work. Using COREQ (or another suitable guideline) will help ensure that key details are transparently reported (e.g., researcher characteristics and reflexivity, sampling strategy, data collection procedures, analysis process, and techniques to support trustworthiness).

Specific comments

1. Provide a concise summary of the original Ajyal Salima intervention components.

- Page 6, line 106 (and/or Introduction, page 4, line 62): Please include a brief, structured description of the core components of the original (non-digital) Ajyal Salima intervention, so readers can understand what is being digitized and what “fidelity” means in this context.

o Also consider a short table/box summarizing details of both versions of the intervention components (e.g., specify target population, setting, key activities/materials, delivery agents, dose/frequency, and hypothesized mechanisms).

2. Map the original and digital intervention components to a behavioral theory.

- To strengthen the conceptual framing of the adaptation process, please consider mapping the intervention components of both the original and the digital Ajyal Salima using a behavioral change framework. For example, the Behavior Change Wheel (BCW [2]), including intervention functions (e.g., education, training, persuasion, enablement, environmental restructuring, modeling).

- This mapping can clarify what functions are preserved, modified, added, or potentially lost in the digitization process, and it can help the reader understand how the platform is expected to influence behaviors.

3. Include Ajyal Salima staff perspectives in the thematic structure to enable comparison.

- Page 26, line 566: please consider presenting staff feedback within the same subthemes used for children, parents, and teachers. This would improve triangulation and allow readers to compare insights across participant groups within the same topics.

- The Results and Discussion sections would benefit from tighter organization and more concise writing. Where possible:

o Group overlapping ideas and avoid repeating participant quotes that make the same point.

4. Clarify next steps for evaluation and scale-up of the digital platform.

- The Discussion would be strengthened by explicitly outlining the next steps for evaluating and/or scaling up the digital Ajyal Salima platform. You may consider using the MRC guidance for developing and evaluating complex interventions [3] to structure this section. For example, describing what should happen next in terms of: further development/refinement of the intervention, feasibility/piloting (including engagement metrics and implementation outcomes), effectiveness evaluation (and potential study designs), and implementation and scale-up considerations.

References

1 Tong, A., Sainsbury, P. & Craig, J. Consolidated criteria for reporting qualitative research (COREQ): a 32-item checklist for interviews and focus groups. International journal for quality in health care : journal of the International Society for Quality in Health Care 19, 349-357, doi:10.1093/intqhc/mzm042 (2007).

2 Michie, S., van Stralen, M. M. & West, R. The behaviour change wheel: a new method for characterising and designing behaviour change interventions. Implementation science : IS 6, 42, doi:10.1186/1748-5908-6-42 (2011).

3 Skivington, K. et al. A new framework for developing and evaluating complex interventions: update of Medical Research Council guidance. BMJ (Clinical research ed.) 374, n2061, doi:10.1136/bmj.n2061 (2021).

**Do you want your identity to be public for this peer review?** For information about this choice, including consent withdrawal, please see our Privacy Policy

Reviewer #1: No

Reviewer #2: No

---

## [Author Response · Author response to Decision Letter 1]

3 Feb 2026

Thank you for your comments please find below a point-by-point answers to your queries, you can also find the amendments in track changes in the manuscript.

Editors:

Answer: we checked that the manuscript meets PLOS ONE’s style requirements.

Answer: The questionnaire on inclusivity has been uploaded

“The intervention was supported by the Nestlé for Healthier Kids initiative – Nestlé Middle East (Grant Number: Award 100 119). The latter had no role in the research design of the study; in the collection, analyses or interpretation of data; in the writing of the manuscript or in the decision to publish the results.”

“Nestle For Healthier Kids Initiative MENA Grant Number: 100119”

Answer: We removed all texts related to the funding from the manuscript. Kindly advise whether the funding statement should appear or not in the Acknowledgement section.

We would like our funding statement to read as follows:

The intervention was supported by the Nestlé for Healthier Kids initiative – Nestlé Middle East (Grant Number: Award 100 119). The latter had no role in the research design of the study; in the collection, analyses or interpretation of data; in the writing of the manuscript or in the decision to publish the results.

Answer: This was added to the cover letter.

Answer: The correct Grant number has been added.

5. Thank* you for stating the following financial disclosure:

“Nestle For Healthier Kids Initiative MENA Grant Number: 100119”

Answer: The amended role of funder was added to the cover letter.

6. Please include a caption for figure 1./

Answer: Caption for figure 1 (which is now figure 2) was added to the manuscript.

7. We note that this data set consists of interview transcripts. Can you please confirm that all participants gave consent for interview transcript to be published?

If they DID provide consent for these transcripts to be published, please also confirm that the transcripts do not contain any potentially identifying information (or let us know if the participants consented to having their personal details published and made publicly available). We consider the following details to be identifying information:

- Names, nicknames, and initials

- Age more specific than round numbers

- GPS coordinates, physical addresses, IP addresses, email addresses

- Information in small sample sizes (e.g. 40 students from X class in X year at X university)

- Specific dates (e.g. visit dates, interview dates)

- ID numbers

Answer: Kindly indicate if by data set you mean the information provided in supplementary table. The information presented in the latter consists of anonymized participant quotes.

Reviewers’ comments:

Reviewer 1:

Below are comments for the authors consideration to improve the manuscript. Title: I suggest removing "Qualitative Study" from the title to improve clarity and impact. If the authors wish to keep it, then additional details on qualitative rigor, including trustworthiness, should be included in the manuscript, and frameworks such as COREQ (Consolidated Criteria for Reporting Qualitative Research) should be provided as a supplementary document

Answer: Thank you for highlighting this matter, we have used the COREQ as the qualitative reporting guideline and it is uploaded as a supplementary file.

Abstract: Line 2-3: ‘Digital technologies are increasingly influencing children’s lives, with many seeking digital platforms for nutritional education.’ This sentence is not clear, does many refer to parents or children? It would be good to start an abstract with a strong sentence. Please include details such as the size and number of the FGDs, type of interview IDI or KII, how many themes? What software was used for the analysis? Clarify also if the FGDs were done separately for parents and children? This information is necessary to be included in the abstract.

Answer: Thank you, we agree that this required further clarification, the sentence has been amended and information regarding the size, number of the FGDs and types of interviews were added.

Materials and Methods: Study participants Which specific interview technique was used? As mentioned in the abstract, provide details on the composition and size of the FGDs and the total number of interviews.

Answer: We appreciate this suggestion and have addressed it to provide further details regarding composition and size of FGGs and interviews. Table 1 was amended accordingly.

What criteria was used for the school selection, and how were the students identified and recruited? How many were reached and actually interviewed?

Answer: Thank you for highlighting this point. School selection criteria included the ability of schools to conduct the intervention as per protocol, the availability of necessary staff, internet access and accessibility to connected devices at students’ homes; students in grades 4 and 5 from each school selected in all countries were reached. 145 were interviewed, and data collection stopped when we reached data saturation. This information has been added to the materials and methods section.

Intervention (Lines 106–116): The description of the intervention lacks details for readers to fully understand the intervention. Consider listing the 10 modules rather than only mentioning the concepts covered, which will give readers an idea of what was implemented. Also include details of the timing within the academic year, platform format (app or web-based), duration per module, sequence of modules, and length of the intervention period. Elaborate on the role of parents and teachers beyond account setup (e.g., supervision, reinforcement, technical support).

Answer: Thank you for raising this point, the intervention section was amended to include additional details on the modules, timeline, duration and platform format. Parents were encouraged to support children’s engagement at home especially in activities with food preparation, this was also addressed in the text.

Data collection Lines 122 –126 – The timeline is unclear. It's mentioned that the intervention lasted around a month, but many students completed the modules in a week or less. Were FGDs and interviews conducted immediately after each participant finished or only after the month-long intervention was completed? Clarifying this is important for understanding recall bias.

Answer: Thank you for this note. Although most students completed the whole program within a week, FGDs and IDIs were conducted with all participants one month following the completion of the intervention. Interview guides and procedures were consistent across countries. This has been amended in the text.

Sample size and Sampling: Provide a rationale for the sample size (this hasn’t been mentioned anywhere) and clarify the sampling strategy, including inclusion and exclusion criteria.

Answer: Thank you for your comment. This qualitative study employed a purposeful, multi stakeholder sampling strategy to capture diverse perspectives on the implementation and impact of the Ajyal Salima program across multiple sociocultural and educational contexts. Inclusion criteria included availability of internet access and accessibility to connected devices at students’ homes. We added this information in the materials and methods section, under the participants part.

Lines 130 – 133 – The phrase “two FGDs per category” is unclear. Please clarify what “category” refers to (e.g., country, grade level, or participant type). Were FGDs conducted separately for each subgroup, or were participants mixed? Indicate whether the FGDs were conducted per country, per class, or across the whole sample, and provide the total number of FGDs held. Also specify session duration (it appears interviews lasted approximately 40 minutes, but was this the same for FGDs?) and whether there were any notable differences in the discussions between grade 4 and grade 5 students.

Answer: Thank you for raising this point. Category refers to participant type. FGDs were conducted separately for each participant type or sub-group and per country. All session lasted around 40 minutes including FGDs, there were no notable differences between grades 4 and 5. This comment was addressed in the manuscript.

Lines 134 – 136- Please state how many teachers were interviewed in total and how many schools they represented. Similarly, report the total number of students and parents who participated in FGDs. This will give readers a clear sense of the sample size and coverage across participating schools. Confirm whether interview guides and procedures were consistent across countries. Data transcription: it's not clear what was used for recording? The phrase ‘auditory means’ is not clear. Who was involved in the translation of the transcripts to English?

Answer: Thank you for your comment, the number of teachers interviewed is now clearly defined in table 1 in the participants section. Interview guides were consistent across countries, digital devices (phone, Ipad.) were used for the recording; bilingual research team members were involved in the translation of transcripts to English. This has been addressed in the manuscript.

Result:

Lines 176 – 177: This information regarding the number of schools and data collection sites is more appropriate for the methods section rather than the results. Please move it accordingly.

Answer: Thank you for your observation. This has been adjusted and it is now moved to the methods section.

Table 1: In Row 5 under “Number of Teachers Interviews,” the entry “3 (4)” seems inconsistent or possibly a typo. Please verify the correct number of interviews and participants.

Answer: Thank you for highlighting this point. The table has been amended to clarify the correct number of interviews and participants.

I suggest adding a table before Table 1 to present the participant profile (children, parents, teachers) in a separate table that summarizes their demographics and distribution for all countries.

Table 1 now clearly presents the participants’ profile as well as their distribution for all countries. Data on participants demographics is not available.

Lines 202 – 207 It is unclear if the quotes listed here are from a single participant or multiple participants. Please clarify and specify the participant type (child, parent, teacher) for each quote to aid transparency and context.

Answer: Thank you for this comment. Those were children’s quotes, this was amended in the text.

Lines 208 – 215 The Results mention that teachers’ focus groups confirmed technical concerns. However, the Methods section states that only interviews were conducted with teachers (not focus groups). Please check this inconsistency. Quotations throughout the Results: It is recommended to indicate the participant type (e.g., Parent, Teacher, Child) and their country at the end of each quote, for example: “Quote text.” (Parent, Palestine); “Quote text.” (Teacher, Jordan); “Quote text.” (Student, Lebanon). Overall, the result section is quite long and too much information is presented, and the authors can reduce this section by reducing the number of quotes, focusing on the key findings, and avoiding unnecessary details such as narrating procedures.

Answer: Thank you for this comment. Teachers’ “focus groups” is amended in the manuscript. The participant type and their country are already specified in the text for the presented quotes. To further enhance clarity, we carefully reviewed the manuscript and added them where the information was missing.

Discussion: Is too long and needs to be reduced, and it's good to consider restructuring the discussion so it follows this logical flow: Overall – in the first paragraph, a summary of the findings needs to be written, and this is a brief restatement of key findings (not repeating the results). Followed by an interpretation of the findings in light of your data, supported by literature, and the final paragraph should be more of the implications for practice and policy. Explain what the findings mean, why they matter, how they relate to existing literature, implications, limitations, and possible future directions.

Some of the paragraphs narrate a general background which not related to the result, and these need to be condensed to make the discussion clear and readable.

Strengths and weaknesses

The Strengths and Limitations section mentions practical points but lacks critical reflection on the study’s methodological strengths and weaknesses. It should better address issues like sampling, data quality, potential biases, and translation processes to enhance transparency and rigor.

Conclusion – needs to be condensed to only highlight the key conclusion based on the study findings and add recommendations. Overall, this manuscript addresses a valuable and timely topic with promising potential for digital nutrition education. With focused revisions to clarify methodology, streamline results, and deepen interpretation, the manuscript will significantly strengthen its contribution to the field

Answer: Thank you for this valuable comment. The discussion has been restructured and reduced as per the comments received. The strengths, weaknesses and conclusion sections have been also amended in the manuscript.

Reviewer 2:

Major comment

1. Use a qualitative reporting guideline (e.g., COREQ1).

- Please select and apply an established reporting guideline for qualitative studies. The COREQ (Consolidated Criteria for Reporting Qualitative Research) checklist would be appropriate for intervie

---

## [Editor Report · Decision Letter 1]

5 Feb 2026

Successes and Challenges of an online based nutrition awareness program in 9–11-year-old children In Four Arab Countries: The Ajyal Salima digital platform Qualitative study

PONE-D-25-25805R1

Dear Dr. Carla Habib-Mourad,

We’re pleased to inform you that your manuscript has been judged scientifically suitable for publication and will be formally accepted for publication once it meets all outstanding technical requirements.

Kind regards,

Nour Amin Elsahoryi, pHD

Academic Editor

PLOS One

Additional Editor Comments (optional):

Thank you for your careful and thorough revision. The manuscript has improved substantially, particularly in terms of clarity of methods, qualitative reporting, and description of the digital intervention. The use of COREQ and the expanded methodological detail strengthen the credibility of the study.

Before final acceptance, please address the following two minor points:

Funding information

Please remove all funding-related text from the manuscript itself (e.g., Acknowledgments or Financial Statement sections). Funding details should be reported only through the journal’s online Funding Statement, as per PLOS ONE policy.

Consent for publication of qualitative data

Please explicitly state whether participants provided consent for the publication of anonymized interview excerpts. If only anonymized quotations are published, clarify this clearly and confirm that no identifying information is included.

Once these points are addressed, the manuscript will be ready for acceptance.

---

## [Editor Report · Acceptance letter]

PONE-D-25-25805R1

PLOS One

Dear Dr. Habib-Mourad,

I'm pleased to inform you that your manuscript has been deemed suitable for publication in PLOS One. Congratulations! Your manuscript is now being handed over to our production team.

Kind regards,

on behalf of

Dr. Nour Amin Elsahoryi

Academic Editor

PLOS One